# Multiple Sulfatase Deficiency from an Ophthalmologist’s Perspective—Case Report and Literature Review

**DOI:** 10.3390/children10030595

**Published:** 2023-03-21

**Authors:** Michael P. Schittkowski, Sabine Naxer, Mohamed Elabbasy, Leonie Herholz, Vivian Breitling, Alan Finglas, Jutta Gärtner, Lars Schlotawa

**Affiliations:** 1Section for Strabismus and Neuroophthalmology, Department of Ophthalmology, University Medical Centre Goettingen; Robert-Koch-Str. 40, 37085 Goettingen, Germany; 2Division for Neuropaediatrics, Department of Paediatrics and Adolescent Medicine, University Medical Centre Goettingen, 37075 Goettingen, Germany; 3MSD Action Foundation, D15 Dublin, Ireland

**Keywords:** multiple sulfatase deficiency (MS), ophthalmology, optical coherence tomography (OCT)

## Abstract

Multiple sulfatase deficiency (MSD) is an extremely rare autosomal recessively inherited disease with a prevalence of 1:500.000 caused by mutations on the sulfatase-modifying-Factor 1 gene (SUMF1). MSD is most specifically characterised by a combination of developmentally retarded psychomotoric functions, neurodegeneration that entails the loss of many already acquired abilities, and by ichthyosis. Other symptoms include those associated with mucopolysaccharidosis, i.e., facial dysmorphy, dwarfism, and hepatosplenomegaly. In 50–75% of all MSD-affected patients, functional or structural ocular damage is likely. MSD seldom affects the anterior segment of the eye. The main pathology these patients present is a highly conspicuous tapetoretinal degeneration, similar to severe Retinitis pigmentosa, that leads to blindness at an early age. An initially five-year-old boy with MSD, genetically verified at his first examination in our opthalmology department (SUMF1 mutations c.776A>T, p.Asn259Ile; c.797A>T, p.Pro266Leu; c.836A>T, p.Ala279Val), and a 4, 5 year regular follow-up are described. The patient had some visual potential (“tunnel view”), which deteriorated dramatically after his fifth birthday. We observed no evidence of worsening retinal involvement in this patient in spite of his progressively worsening clinical symptoms, extending to total blindness/no light perception. OCT revealed that the outer retinal layers containing photoreceptors were diseased; the ellipsoid zone was only partially discernible and the outer nuclear layer appeared to be thinned out. The inner nuclear layer, ganglion cell layer, and retinal nerve fibre layer were indistinguishable. These anomalies are indicative of a severe pathology within the retina’s inner layers. Characteristic anomalies in the fundus should stimulate clinicians to suspect a case of MSD in their differential diagnosis, and thus to order thorough genetic and paediatric diagnostics.

## 1. Introduction

Multiple sulfatase deficiency (MSD) is an extremely rare autosomal recessively inherited disease with a prevalence of 1:500.000 [1]. Its disease course is poorly understood, as only 150 patients with MSD have been reported in the literature [2]. It is caused by mutations on the sulfatase-modifying-Factor 1 gene (SUMF1), which is coded for the formylglycin-generating enzyme (FGE). FGE is essential to post-translationally activate all newly synthesised sulfatases in the endoplasmic reticulum. SUMF1 mutations trigger the FGE’s loss of function, and that impairment in turn damages all sulfatase activity. Sulfatases are necessary for the hydrolysis of glycosaminoglycans, sulfolipids, and steroid hormones. Of the human genome’s 17 sulfatases, 11 are located in lysosomes. Isolated sulfatase defects cause diseases such as metachromatic leukodystrophy (MLD, MIM #250100), X-chromosomal ichthyosis (XLI, #308100), Morquio A Syndrome (MPS IVA, MIM #), Sanfilippo Syndromes A and D (MPSIIIA, MIM #252900, MPSIIID, MIM #252940), and Hunter’s Syndrome (MPSII, MIM #309900). 

As all sulfatases are impaired in MSD, its clinical presentation reveals a variety of symptoms depending on which sulfatase is deficient. MSD is most specifically characterised by a combination of developmentally retarded psychomotoric functions, neurodegeneration that entails the loss of many already acquired abilities, and by ichthyosis. Other symptoms include those associated with mucopolysaccharidosis, i.e., facial dysmorphia, dwarfism, and hepatosplenomegaly. 

The disease’s course is progressive, and its spectrum encompasses severe as well as mild forms that depend on the FGE protein’s remaining function; the course varies in conjunction with the time at which symptoms appear. There is currently no causal therapy for MSD. The 50% survival rate is about 13 years [1]. Pulmonary infections were the most frequent cause for an early death in MSD children [1,2].

Individual case reports addressing MSD have described pathological ocular anomalies [3,4,5,6]. A recent systematic review and meta-analysis of MSD case reports revealed that in 50–75% of all MSD-affected patients, functional or structural ocular damage is likely [1,2]. Mutations or deficits on the ARS gene cause lysosomal-storage diseases such as mucopolysaccharidosis (MPS) and metachromatic leukodystrophy which, in affected individuals, often lead to symptoms of retinal dystrophy resembling the retinal pigment anomalies associated with retinitis pigmentosa (RP) [7].

Below, we describe in detail our ophthalmological findings after regularly following up on an MSD patient over a four-and-a-half-year period, for the first time including retinal OCT diagnostics. 

## 2. Case Report

We describe the condition of a five-year-old boy with genetically verified MSD at his first examination in our opthalmology clinic at the University Medical Center in Goettingen (SUMF1 mutations c.776A>T, p.Asn259Ile; c.797A>T, p.Pro266Leu; c.836A>T, p.Ala279Val).

While recording the boy’s medical history, the boy’s father reported that his son had been able to visually recognise his parents until about age 4, but also that they needed to place their faces directly in front of him. The boy could not recognise them from lateral perspectives in his visual field. The parents referred to this as “tunnel vision”. The parents stated that the boy could throw a football with a reasonable degree of accuracy back to the parent up until his third birthday, at least. They could not say when exactly this “tunnel vision” began during their child’s fourth year. 

The boy’s vision deteriorated dramatically after his fifth birthday. He was only able to react to light after a long, adaptive exposure to darkness that was suddenly interrupted by exposure to a bright light, i.e., in a darkened bedroom when the door is quietly opened and bright light from the hallway enters the room. 

The boy underwent his first ophthalmological exam in our institute at age five and a half; we examined him again after 29 (age 7 y, 11 mo), 33 (8 y, 3 mo), 37 (8 y, 7 mo), 50 (9 y, 8 mo), and 55 months (10 y, 1 mo) after his first presentation. 

At each examination, we observed dilated, round, isochore pupils. Their reaction to light was already extremely delayed and weak at the first presentation. Despite repeated tests, the boy was unable to fixate on objects or light. We were also unable to trigger an optokinetic nystagmus. 

His pupils’ extremely sluggish reaction to light which we observed initially became even weaker at follow-up months 29, 33, and 37 after his first presentation. From the timepoint of his exam at the end of his 10th year (50 months after his baseline exam), no pupil reaction remained whatsoever; even the dazzling exam light we exposed him to triggered no defensive reaction. 

His ophthalmological exams consistently revealed normal, inconspicuous eyelids, with a normal position and mobility. The optic media were unremarkable after hand-held slit lamp examination; note that we detected no sign of corneal opacity or cataract. Binocular funduscopy revealed severe pathological anomalies that entailed bilateral pale optic nerve heads with sharp margins (Figure 1). The macula had neither a foveal central nor a wall reflex, and the fovea showed ophthalmoscopically loosened structures with some degree of hypopigmentation. The vessels, especially the arteries, were extremely constricted; in the nasal direction they were barely discernible, and they did not extend into the periphery. We observed pronounced segmental pigment loss nasally and temporally, mainly outside the temporal vascular arcade. An extremely translucent choroid was seen in several areas (Figure 1). There were bone spicules in the mid-periphery, especially in the nasal and lower nasal position (Figure 1). Neither our (often barely possible) photo documentation nor the ophthalmoscopic evidence yielded any indication of the disease’s progress. (Figure 1).

We re-examined the boy shortly before his eighth birthday and finally managed (after several failed attempts) to carry out a retinal OCT exam (Figure 2). Here, the retinal pigment epithelia (RPE) is readily identifiable. The choroid is more obvious than usual, as is characteristic of the advanced retinal dystrophy in RP. The outer retinal layers containing photoreceptors are diseased; the ellipsoid zone is only partially discernible, and the ONL (outer nuclear layer), which contains the photoreceptors’ nuclei, appears to be thinned out. The INL (inner nuclear layer), GCL (ganglion cell layer), and RNFL (retinal nerve fibre layer) were indistinguishable. These anomalies are indicative of a severe pathology within the retina’s inner layers. 

We were unable to photograph the foveal structure or to perform pRNFL-OCT because of a lack of co-operation. A scan of the left eye (no figure) revealed a similar condition but in technically worse quality.

## 3. Discussion

MSD is an extremely rare disease. Only about 150 cases thereof have been reported in the literature [1,2]. MSD’s very low prevalence, its devastating symptoms, and the brief lifespan of those suffering from the disease are probable reasons for the paucity of published ophthalmological data on MSD. This lack of systematic research is attributable to MSD’s rarity.

In their review published in 1984, Bateman et al. [3] described 31 cases, including case reports referring to absolutely no, or only incomplete, ophthalmological data. In neither a recent systematic review addressing MSD’s natural course, nor a meta-analysis of all published cases until 2020, are any specific data provided on ophthalmological symptoms [1]. There is a 70–80% frequency of pathological findings after the ophthalmological examination of MSD patients [2]. Descriptions of individual findings, as in the present case history, provide the sole source of information on the ocular anomalies that are characteristic of MSD. 

It is no wonder that the aforementioned severe structural anomalies cause the extreme functional visual impairments that ophthalmologists encounter in these patients [3]. The severe visual disabilities that MSD patients face make it very difficult or impossible for ophthalmologists to carry out functional investigations (of the visual acuity and/or visual field). Our close observation of this patient and our taking a thorough patient history are thus even more significant. 

Our patient apparently had very little vision left already during his fourth year of life; he was only capable of perceiving his parents when they held their heads right in front of his face. In that case, we can assume his visual acuity (VA) amounted to less than 0.1 according to the German system. The “tunnel vision” his parents described indicated a central residual visual field that disappeared in the years thereafter. The boy lost his ability to perceive light entirely during his 9th and 10th years of life. We cannot say (because of the lack of findings) whether his visual development during the first two to three years was physiologically normal. 

During our nearly five-year observation of this boy, we identified no structural anomalies in the anterior eye segment, especially the cornea and lens; such anomalies have likewise seldom been reported in conjunction with MSD in the literature [1]. Published reports refer to one case of lens involvement [3] and one of corneal involvement [8], which would indicate a prevalence of roughly 0.7%. 

Bateman et al. [3] described their slit lamp microscopic findings in a patient aged 14 years with continuous bilateral “grey-velvety” opacity in the anterior lens capsule’s periphery. As that was irrelevant for visual acuity, they described no therapy or disease progress. Vamos et al. [8] published their case history of a 2½-month-old infant boy with bilateral “corneal clouding” (during slit lamp examination), but they did not describe it more in detail. 

Corneal anomalies are generally associated with various mucopolysaccharidoses (mainly MPS I, IV, VII). In one of the first published case reports, Cogan et al. [9] implied an MSD, but it was more likely an MPS I or something at least associated with it.

There is a single case report of albinism in association with a type of photophobia, the transilluminability of an iris, and foveal hypoplasia coinciding with a genetically proven MSD [10], but no data are provided on the peripheral retina. A relationship between albinism and MSD is unlikely.

MSD’s typical structural ophthalmological anomalies are observed in the *fundus*. As with our patient, others (when diagnosed with MSD) reveal very advanced optic atrophy. Depending on the age of diagnosis and the disease stage, the optic disc usually stands out as bright yellow to white, with sharp margins.

Our patient’s retinal periphery contained hyperpigmentation in the form of bone spicules (Figure 2) present only in sectors, not circular. His (‘bone spicules’) were very densely packed, reminding us of the clinical manifestation of retinitis pigmentosa (RP). Such similarity to RP is the most frequently reported clinical evidence in the posterior eye [5,11,12,13,14].

There are a few reports of only mild anomalies in the pigment epithelium (minor depigmentation) [3,12,14]. We suspect that such data indicate the quantitative extent of MSD’s various degrees of severity. Increasing numbers of bone spicules seem to be generally associated with a progressing disease course, extending in their distribution closer to the posterior pole.

Histologically speaking, bone spicules are pigment epithelia that have proliferated perivascularly [15]. They constitute a reactive pattern of the retinal pigment epithelium to genetic (i.e., RP syndromes), mechanic (retinopathia sclopetaria after a ocular contusion), inflammatory (chorioretinitis), ischaemic (choroidal vessel occlusion), or toxic (i.e., as a side effect of chlorpromazine) stimuli [15]. 

Concomitant with such severe retinal and pigment epithelial anomalies are typically constricted vessels, as is characteristic of other tapetoretinal diseases. Individual sulfatase defects such as various MPS diseases and MLD also reveal frequent indications of retinal dystrophy, as is usually observed in patients with RP [7]. As not every genetic sulfatase defect has been correlated with a specific disease manifestation, not yet clarified MSD symptoms may be related to the loss of function of corresponding sulfatases. There is recent evidence that Usher Syndrome Type IV correlates with an arylsulfatase G deficiency. These patients also suffer from RP and progressive vision loss [16]. Half of the patients with a recently proven new MPS disease (MPS 10) have an arylsulfatase K defect; however, they exhibit comparatively little eye involvement, such as minor lens and vitreous body opacity, as well as mild retinal pigmentation temporal of the fovea [17].

With this case report, we are, to best of our knowledge, the first to present retinal OCT findings in an MSD patient. What stood out in this patient was the retina’s extreme structural disintegration. It would be fascinating to be able to examine this disease’s clinical symptoms and manifestations as it progresses, in order to discover any correlations. However, as OCT exams require a minimum of co-operation, they are infeasible without adequate compliance, or, as in our case, are severely limited. To have examined this child under sedation and with a hand-held unit did not seem ethically justifiable to us. 

We observed no evidence of worsening retinal involvement in this patient in spite of his progressively worsening clinical symptoms, which extended to total blindness/no light perception. We assume that this is evidence that the boy’s loss of visual function was nearly finalised by the time we first examined him at age 5½ years; that is, most of the essential dynamics of this disease process were already behind him. To circumvent this problem, affected children would need to be ophthalmologically examined as early as possible. 

## 4. Conclusions

MSD is an extremely rare disease entity that seldom affects the anterior segment of the eye. The main pathology these patients present is a highly conspicuous tapetoretinal degeneration, similar to severe RP, that leads to blindness at an early age. As there is still no therapy for it, MSD diagnostics in severely affected children should be limited to funduscopy to assess the optic nerve head and the search for bone spicules. We will have to wait for additional evidence from future OCT examinations. The characteristic anomalies in the fundus should stimulate clinicians to suspect a case of MSD in their differential diagnosis, and thus to order thorough genetic and paediatric diagnostics.

## Figures and Tables

**Figure 1 children-10-00595-f001:**
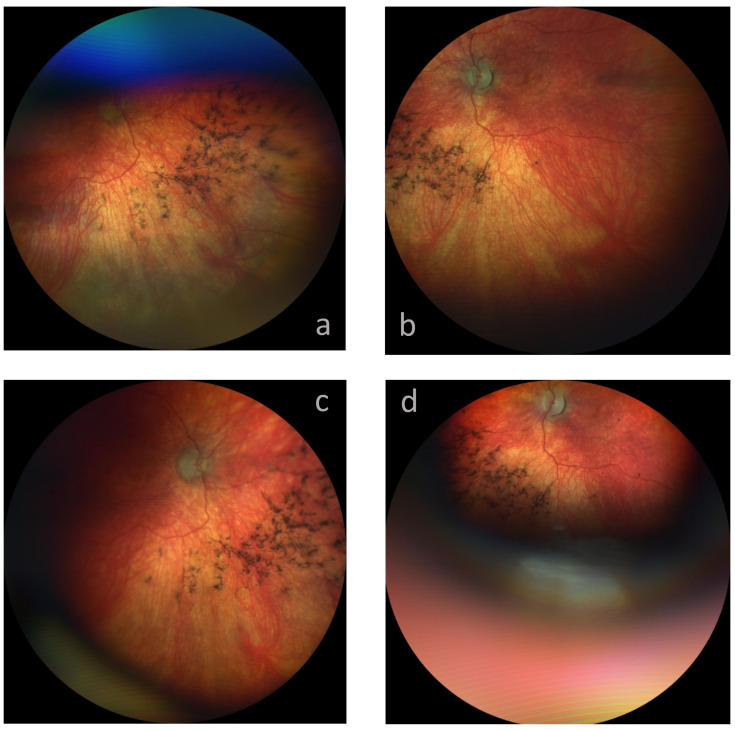
Fundus photograph of the right (**a**,**c**) and left (**b**,**d**) eyes, above at age 7 years and 11 months (**a**,**b**), below 21 months later (**c**,**d**): despite the images’ compromised quality, note the consistency of the pigment anomalies.

**Figure 2 children-10-00595-f002:**
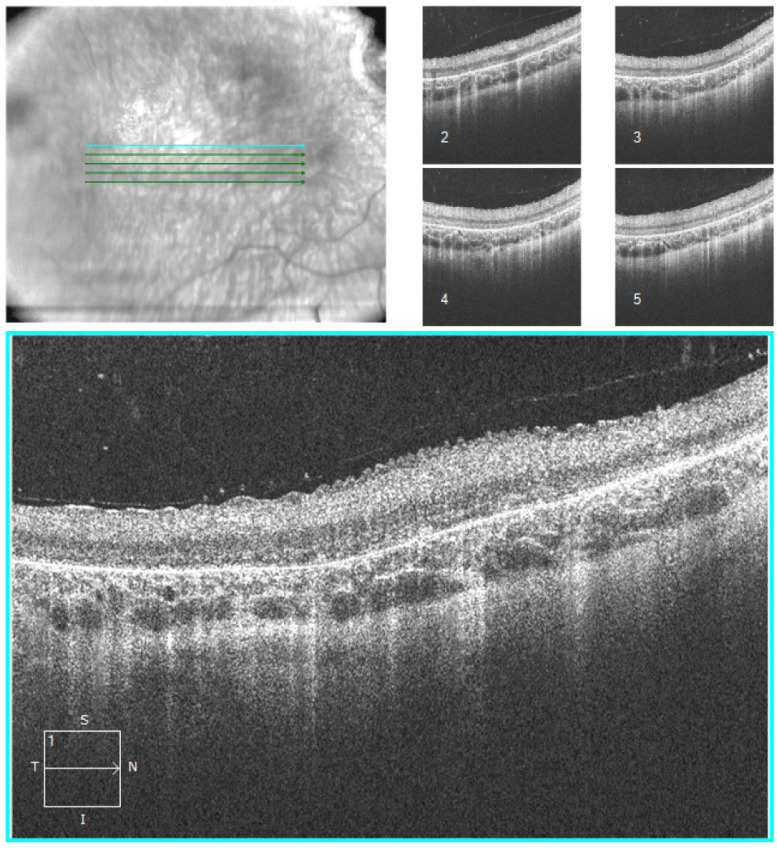
Macula-OCT (Carl Zeiss Meditec) of the right eye (scan-angle 0°, distance 0.25 mm, length 6 mm): severe structural disorganisation in the retina (details s. text).S-superior, n-nasally, I-inferior, T-temporally.

## Data Availability

Not applicable.

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
