# Peer review of "Multiple Sulfatase Deficiency from an Ophthalmologist’s Perspective—Case Report and Literature Review"

_children, 2023, doi:10.3390/children10030595_

Round 1

Reviewer 1 Report

I congratulate the authors for this very interesting and well documented case report of a rare disease.

The figures are informative and the case well presented with a very interesting discussion and review of the literature. 

My only concerns are the numerous typo errors and grammar mistakes.

The whole manuscript should be revised in terms of syntax and grammar.

For example:

Line 91: The sentence should be "His pupils extremely sluggish reaction to light". Authors should delete "were"

Line 98: "The ocular fundus revealed severe pathological anomalies entailing normal, pale, bilateral optic nerve heads". How can optic nerve head be "pale" and "normal"??

Line 104: The following sentenc should be rearranged "exceeding beyond above and below it and as far as the papilla an extremely translucent choroid in several areas."

Line 164 : "but they do not (should be "did not") describe it more detail."

Line 177: They should add "His ('bone spicules') were very densely packed"

Line 180: The following sentence should be rearranged "There are a few reports of anomalies in the pigment epithelium or as minor depigmentation."

And there many other typo-errors that should checked and corrected.

Author Response

Reviewer I

I congratulate the authors for this very interesting and well documented case report of a rare disease. The figures are informative and the case well presented with a very interesting discussion and review of the literature.

 We thank the reviewer very much for this positive assessment and for the precious comments.

My only concerns are the numerous typo errors and grammar mistakes. The whole manuscript should be revised in terms of syntax and grammar.

 please see below

For example:

Line 91: The sentence should be "His pupils extremely sluggish reaction to light". Authors should delete "were"

 was done this way

Line 98: "The ocular fundus revealed severe pathological anomalies entailing normal, pale, bilateral optic nerve heads". How can optic nerve head be "pale" and "normal"??

 We are very sorry, of course the optic disc was not “normal”, please note the changes.

Line 104: The following sentenc should be rearranged "exceeding beyond above and below it and as far as the papilla an extremely translucent choroid in several areas."

 The somewhat confusing wording has been changed.

Line 164 : "but they do not (should be "did not") describe it more detail."

 was done this way

Line 177: They should add "His ('bone spicules') were very densely packed"

 was done this way

Line 180: The following sentence should be rearranged "There are a few reports of anomalies in the pigment epithelium or as minor depigmentation."

 The somewhat confusing wording has been changed as well.

And there many other typo-errors that should checked and corrected.

 The manuscript has been reviewed again and numerous inconsistencies for which the authors apologize have been changed. All changes are marked in review mode.

Reviewer 2 Report

The authors presented a case report describing the ophthalmological characteristics of MSD.

The manuscript is with merit and worth reporting, some minor comments to please be addressed are the following:

- in the methods the authors should add the location of the hospital where the ophthalmological visits occurred

- the authors should add a sentence indicating if the consent acquired from the parents of the patients to publish this case (imaging)

- was the intra-ocular pressure measured? Do patients with MSD have fish of child glaucoma?

- was the ONH examined at the slit lamp/OCT?

- in general, the authors should add information about life expectancy and death causes for patients with MSD

- please revise the use of abbreviations in the manuscript and be sure that each time an abbreviations is used the corresposponding explanation has been provided

Author Response

Reviewer II

The authors presented a case report describing the ophthalmological characteristics of MSD. The manuscript is with merit and worth reporting, some minor comments to please be addressed are the following:

 We thank the reviewer very much for this positive assessment and for the valuable comments.

- in the methods the authors should add the location of the hospital where the ophthalmological visits occurred

 was done this way (l. 74)

- the authors should add a sentence indicating if the consent acquired from the parents of the patients to publish this case (imaging)

 was already partly included, was still supplemented (l. 263)

- was the intra-ocular pressure measured? Do patients with MSD have fish of child glaucoma?

 The intraocular pressure could not be measured due to outlined compliance problems. Glaucoma is not a known problem in MSD.

- was the ONH examined at the slit lamp/OCT?

 see additions (l. 107, 108, 144)

- in general, the authors should add information about life expectancy and death causes for patients with MSD

 was already partly included, was still supplemented (l. 59-61)

- please revise the use of abbreviations in the manuscript and be sure that each time an abbreviation is used the corresposponding explanation has been provided

 has been implemented (see l. 215 and following
